# The bacterial multidrug resistance regulator BmrR distorts promoter DNA to activate transcription

Chengli Fang [1,2,8], Linyu Li[3,8], Yihan Zhao[4], Xiaoxian Wu[1,2], Steven J. Philips[5], Linlin You[1,2], Mingkang Zhong[3], Xiaojin Shi[3], Thomas V. O'Halloran [5,6,7], Qunyi Li[3✉] & Yu Zhang [1✉]

The MerR-family proteins represent a unique family of bacteria transcription factors (TFs), which activate transcription in a manner distinct from canonical ones. Here, we report a cryo-EM structure of a *B. subtilis* transcription activation complex comprising *B. subtilis* six-subunit ($2\alpha\beta\beta'\omega\varepsilon$) RNA Polymerase (RNAP) core enzyme, $\sigma^A$, a promoter DNA, and the ligand-bound *B. subtilis* BmrR, a prototype of MerR-family TFs. The structure reveals that RNAP and BmrR recognize the upstream promoter DNA from opposite faces and induce four significant kinks from the $-35$ element to the $-10$ element of the promoter DNA in a cooperative manner, which restores otherwise inactive promoter activity by shortening the length of promoter non-optimal $-35/-10$ spacer. Our structure supports a DNA-distortion and RNAP-non-contact paradigm of transcriptional activation by MerR TFs.

[1] Key Laboratory of Synthetic Biology, CAS Center for Excellence in Molecular Plant Sciences, Chinese Academy of Sciences, 200032 Shanghai, China. [2] University of Chinese Academy of Sciences, 100049 Beijing, China. [3] Clinical Pharmacy Laboratory, Huashan Hospital, Fudan University, 200040 Shanghai, China. [4] Key Laboratory of Plant Stress Biology, State Key Laboratory of Cotton Biology, School of Life Sciences, Henan University, 475004 Kaifeng, China. [5] Department of Chemistry, Northwestern University, Evanston, IL 60208, USA. [6] Department of Molecular Biosciences, Northwestern University, Evanston, IL 60208, USA. [7] The Chemistry of Life Processes Institute, Northwestern University, Evanston, IL 60208, USA. [8] These authors contributed equally: Chengli Fang, Linyu Li. ✉email: liqunyi@huashan.org.cn; yzhang@sippe.ac.cn

Transcription activation involves cooperative interplay among RNA polymerase (RNAP) holoenzyme, promoter DNA, and transcription factors (TFs)[1,2]. The prevailing "recruitment" model of bacterial transcription activation specifies that TFs enrich RNAP at specific genomic loci by making direct interactions with both RNAP and promoter DNA, thereby creating additional linkage between them[3]. In line with their function, canonical bacterial TFs are typically equipped with a DNA-binding domain (DBD), which reads specific sequence in the form of double-stranded DNA (dsDNA), and small surface patches, which interact with the RNAP-α subunit or domain R4 of σ factors ($\sigma_4$)[4,5].

Previous biochemical and structural evidence suggests that the MerR-family TFs probably activate transcription in a manner distinct from the canonical TFs[6–12]. The MerR-family TFs, based on protein domain architecture, are divided into two subgroups. The first subgroup is composed of metal- or oxidative stress-responsive factors, which are ultrasensitive to cellular metal ions, such as *Escherichia coli* CueR[13], *E. coli* ZntR[14], *Ralstonia metallidurans* PbrR[15], *Pseudomonas putida* CadR[16], and *Synchocystis* PCC 6803 CoaR[17], or to cellular superoxide, such as *E. coli* SoxR[18]. The second subgroup is composed of multidrug-responsive regulators, which are capable of binding multiple compounds with diverse chemical properties, such as *Bacillus subtilis* BmrR[19], *B. subtilis* BltR[20], and *B. subtilis* Mta[21]. MerR-TFs recognize a long palindromic operator that locates in a region completely overlapping with the spacer between the −35 and −10 elements (two key regions of promoter DNA for RNAP recognition) of their regulated promoter DNA[1,6,8,21].

The reported crystal structures of binary complexes of MerR-family TFs bound with their cognate operator DNA revealed a striking distortion around the center of their palindromic operators[7,8,12,18]. Given that the MerR-TF-regulated promoters contain an unusually long 19-/20-bp spacer compared with the regular promoters with an optimal 17-bp spacer between the −35 and −10 elements[1,6,8,21], such DNA distortion was proposed to realign the geometry of the non-optimal promoter DNA and to restore the promoter activity[7,9,22]. However, due to the lack of RNAP in those crystal structures, it is unknown how MerR-TFs manage to reconcile with RNAP to bind and reshape promoter DNA with near-complete spatial overlapping binding site for RNAP (spanning from −40 to +20) and MerR-family TFs (spanning from −35 to −15).

*Bacillus subtilis* BmrR is the prototype of the second subgroup of MerR-TFs[7,23–25]. It regulates the expression of *Bmr* (bacterial multidrug resistance), a bacterial efflux pump recognizing cationic amphiphilic chemicals[19]. Similar to the substrate spectrum of the Bmr efflux pump, BmrR is capable of recognizing xenobiotic cationic chemicals with a broad range of structural, chemical, and binding properties[24,26–28]. A large collection of the reported crystal structure of BmrR apoprotein and BmrR complexed with drug and DNA reveal substantial conformational dynamics of BmrR[7,23–25,29,30], thereby providing a foundation to explore the conformation change upon RNAP recruitment and the underlying transcription activation mechanism of MerR-TFs. In the present study, we report a cryo-EM structure of transcription activation complex (TAC) with *B. subtilis* BmrR at 4.4 Å. The structure shows that RNAP and BmrR bind at the opposite faces of the promoter DNA. Albeit barely interacting with each other, RNAP and BmrR together induce four kinks at the upstream promoter dsDNA, which shortens the non-optimal −35/−10 spacer for efficient promoter unwinding. Our work supports an RNAP-non-contact transcription activation paradigm only involving DNA distortion.

## Results

**The overall structure of BmrR-TAC.** To assemble the BmrR-TAC, we purified endogenous *B. subtilis* RNAP-$\sigma^A$ holoenzyme and recombinant *B. subtilis* BmrR (Supplementary Fig. 1). We confirmed that BmrR is able to activate p*bmr* in the presence of tetraphenylphosphonium (TPP) (Supplementary Fig. 1c). Bacterial TACs and transcription initiation complexes assembled using the pre-melted promoter DNA exhibited almost identical conformation and interactions between protein and DNA as the complexes obtained using duplex promoter DNA[31–34]. Therefore, to improve the sample homogeneity and enhance protein–DNA interaction, we designed a P*bmr* chimeric promoter DNA comprising a 27-bp (−38 to −12) upstream promoter dsDNA of the wild-type p*bmr* DNA, a 13-bp (−11 to +2) non-complementary transcription bubble with consensus sequence at the −10 element, the discriminator element, and the core-recognition element, and a 10-bp (+3 to +12) downstream promoter dsDNA with G/C-rich sequences (Fig. 1a and Supplementary Fig. 1a). BmrR is capable of promoting RNAP-promoter DNA open complex (RPo) formation of both the wild-type and chimeric P*bmr* promoters (Supplementary Fig. 1d).

The catalytically competent BmrR-TAC was reconstituted using the *Bs* RNAP holoenzyme, TPP-bound *Bs*-BmrR, and the above-mentioned nucleic-acid scaffold (Supplementary Fig. 1e±h).

The final cryo-electron microscopy (cryo-EM) map, reconstructed using a total of 103,226 single particles, was refined to a nominal resolution of 4.4 Å (Supplementary Fig. 2a–c), with ~4 Å at the center of RNAP and ~6 Å at the peripheral BmrR (Supplementary Fig. 2d). The cryo-EM map shows a clear signal for BmrR dimer, 50-bp promoter DNA (−38 to +12) with the exception of disordered template ssDNA (−11 to +2), five major domains of *B. subtilis* $\sigma^A$ subunit ($\sigma^A_{1.2}$, $\sigma^A_2$, $\sigma^A_{3.1}$, $\sigma^A_{3.2}$, and $\sigma^A_4$), and six subunits of *Bs* RNAP (two α subunit, one β subunit, one β′ subunit, and one ω subunit, as well as the ε subunit) (Fig. 1 and Supplementary Fig. 3a, b).

In addition to the universally conserved five subunits ($2\alpha\beta\beta'\omega$), *B. subtilis* RNAP contains two additional small subunits, one of which is the RNAP-ε subunit, encoded by *B. subtilis rpoY* and broadly present in phylum *Firmicute*[35]. Although it was identified 40 years ago[36], it is still unknown how it interacts with the rest of RNAP core enzyme. Our cryo-EM map of *B. subtilis* BmrR-TAC structure shows a clear signal for the RNAP-ε subunit (Fig. 1f), which locates in a cavity created by RNAP-α, -β, and -β′ subunits at the base of RNAP core enzyme, a location far away from the RNAP active-center cleft and distinct from that of its structurally related phage protein T7 GP2[36,37]. Although no defect of transcription activity and bacteria growth was observed in the absence of RNAP-ε subunit[36], the RNAP-ε subunit makes extensive interactions with the RNAP-α, -β, and -β′ subunits (Supplementary Fig. 3), suggesting that it might be involved in stability or assembly of RNAP core enzyme.

**Cooperative recognition of promoter DNA by BmrR and RNAP.** In the BmrR-TAC structure, the entire promoter DNA was protected by RNAP and BmrR dimer (Fig. 2a). The *B. subtilis* BmrR-TAC structure shows a closed-clamp conformation of RNAP and properly engaged transcription bubble and downstream dsDNA in the RNAP main cleft as previously reported RPo structures (Supplementary Fig. 4a–e)[31,32].

BmrR dimerizes and makes interactions with its operator DNA essentially as in previously reported crystal structure of BmrR–DNA complexes[7,24], suggesting that RNAP engagement induces neither significant conformational change to BmrR dimer nor changes of the interaction between BmrR dimer and its operator DNA (Fig. 2b, c). Briefly, the two BmrR protomers

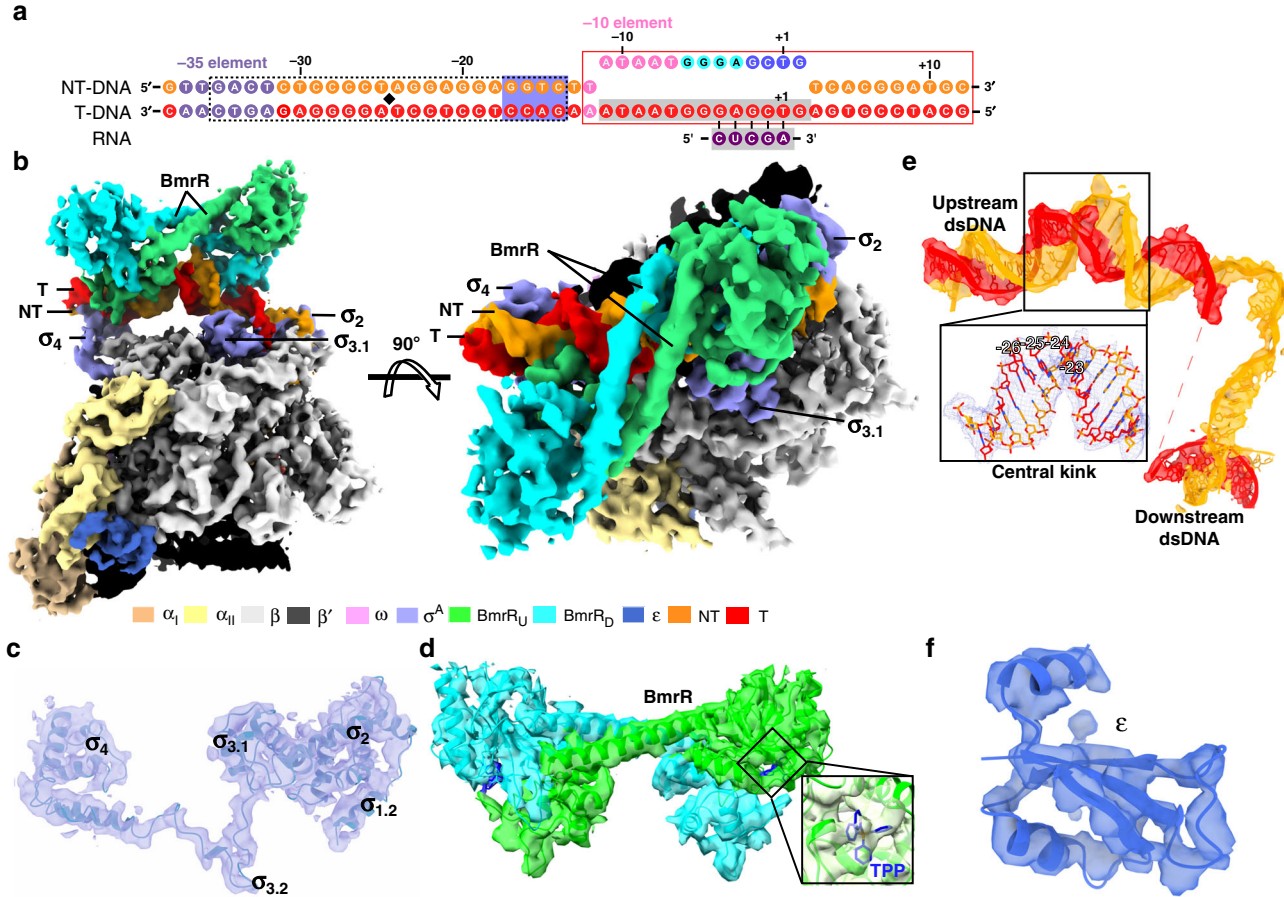

**Fig. 1 The cryo-EM map and model of *B. subtilis* BmrR transcription activation complex (BmrR-TAC). a** Nucleic acid scaffold used for cryo-EM structure determination of BmrR-TAC. Black diamond, the center of BmrR operator; orange, non-template DNA; red, template DNA; light blue, −35 element; pink, −10 element; cyan, discriminator element; blue, core-recognition element; purple, RNA; dashed box, BmrR operator DNA; gray shaded box, disordered nucleotides in the structure; red box, sequenced modified for improving complex stability; and blue shaded box, the extended −10 region. **b** The top and front view-orientations of cryo-EM map of *B. subtilis* BmrR-TAC. The RNAP subunits, nucleic acids, and BmrR are colored as in the color scheme. **c** The cryo-EM map and model for *B. subtilis* σ$^A$. **d** The cryo-EM map and model for *B. subtilis* BmrR dimer. The box highlights the map for tetraphenylphosphonium (TPP) in the BmrR-LBD. **e** The cryo-EM map and model for the promoter DNA. The inset box highlights the promoter central kink. **f** The cryo-EM map for *B. subtilis* RNAP-ε subunit. The cryo-EM maps were contoured at 0.00924 and structure figures were prepared in Chimera X.

(BmrR$_{US}$ and BmrR$_{DS}$) were zippered together through the central helices (residues 77–115) of two protomers and further glued by an interaction between the DBD (BmrR-DBD) of one protomer and the ligand-binding domain (BmrR-LBD) of the other protomer (Fig. 2b). The central zipper helix adopts a straight conformation, as observed in the crystal structure of BmrR–DNA complexes (Supplementary Fig. 5a)[7]. Although the current resolution does not permit unambiguous modeling of the small-molecular ligand TPP, the cryo-EM map shows a strong signal in the ligand-binding site identical to the location of tetraphenylantimonium (TPSb, an analog of TPP) in the crystal structure of TPSb-bound BmrR[7], suggesting presence of the compound in the structure (Fig. 1d; insertion box). The BmrR-DBD of each protomer adopts a winged helix–turn–helix (wHTH) fold and contacts the major groove and minor groove using its HTH and wing loop motifs, respectively (Fig. 3a, b).

Although the −35 element and the upstream half of BmrR operator (BmrR-O$_{US}$; −36 to −25) are partially overlapped, the first three positions (−37 to −35) of the −35 element retain consensus sequence (5′-TTG-3′) for most of the BmrR-regulated promoters (Fig. 1a and Supplementary Table 2), and σ$^A_4$ recognizes the three nucleotides of the −35 element in BmrR-TAC in a manner essentially the same as it does in the crystal

structure of σ$^A_4$/−35 DNA[38] (Supplementary Fig. 4d). The wing loop of BmrR$_{US}$-DBD inserts into the minor groove of the last three position (−34 to −32) of the −35 element and recognizes its operator sequence without interfering with the interaction between σ$^A_4$ and promoter DNA (Fig. 3a, b). Such mode of interaction allows BmrR$_{US}$ and σ$^A_4$ recognizing their respective sequences around the −35 region from the opposite faces of promoter DNA involving no protein–protein interaction (Fig. 3b). This architecture of BmrR-TAC is in sharp contrast to the assembly of bacterial TACs with class II transcription TFs, which make extensive interaction with σ$^A_4$ and RNAP-α subunit, overtake the −35 element of promoter DNA, and result in a loosed interaction of σ$^A_4$ on the degenerate −35 element[4].

The downstream BmrR operator BmrR-O$_{DS}$ completely overlaps with the extended −10 region of the promoter DNA (Fig. 1a). The BmrR$_{DS}$-DBD recognizes its operator DNA by placing the recognition helix and wing loop into the major and minor grooves on the same face of the promoter DNA, respectively, in a manner that allowing the σ$^A_{3.1}$ to access the major groove of the promoter DNA from the opposite face in a sequence-independent fashion (Fig. 3c). Similar to BmrR$_{US}$, BmrR$_{DS}$ makes no interaction with either σ$^A$ or RNAP core enzyme (Fig. 3d). In the promoter regions of downstream BmrR

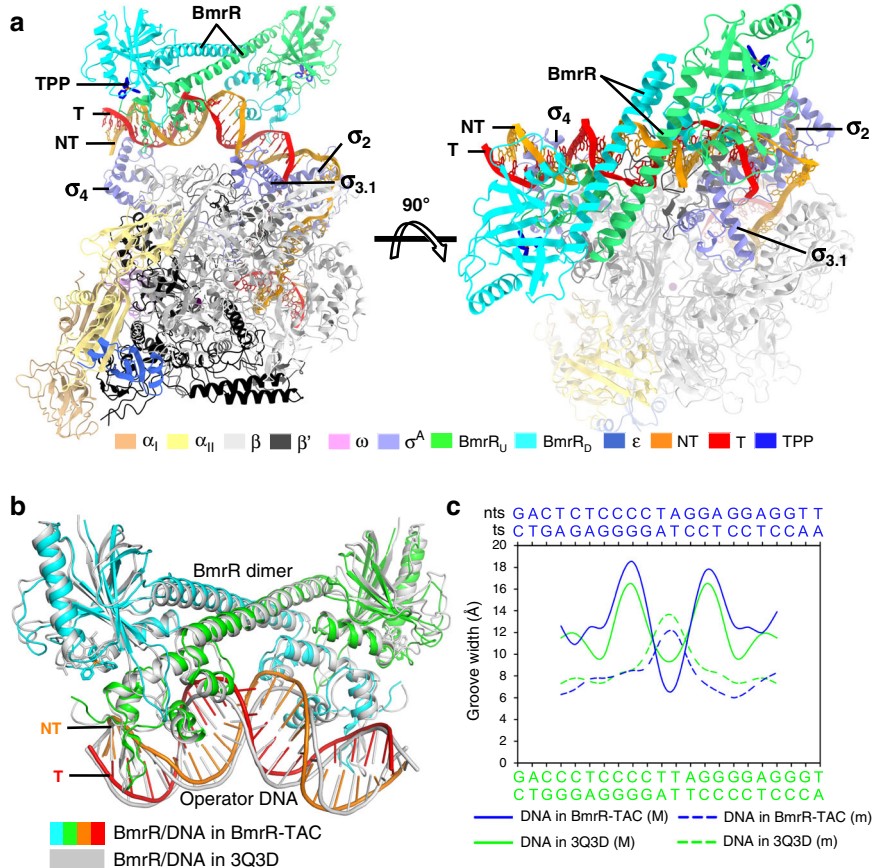

**Fig. 2 The overall structure of BmrR-TAC. a** The top and front view-orientations of the BmrR-TAC structure. **b** Structure superimposition of BmrR/DNA in BmrR-TAC and BmrR/DNA in the crystal structure of a puromycin-bound BmrR/DNA complex (PDB: 3Q3D). **c** Major (M) and minor (m) groove widths of BmrR operator dsDNA in the BmrR-TAC structure and BmrR/DNA binary complex.

operator, the *Bs* RNAP-σ$^A$ holoenzyme unwinds the −10 element and accommodates the transcription bubble and downstream dsDNA in a similar manner as other bacterial RNAP (Supplementary Fig. 4b–e)[31,32,39].

**BmrR and RNAP induce distortion of upstream promoter DNA**. The upstream promoter dsDNA is significantly distorted by BmrR and σ$^A$ compared with the near straight upstream dsDNA in *E. coli* RPo (Fig. 4a, b and Supplementary Fig. 4f). We observed a large kink (kink 3) at the center of its palindromic operator DNA (−24) accompanied by a widened minor groove and a narrowed major groove around the unpaired central two A: T base pairs (Figs. 1e and 2c)[7]. Notably, such kink causes a 90° sharp turn of the helical axis of the upstream promoter dsDNA (Fig. 4a), similar to the TBP-distorted TATA box in eukaryotic transcription pre-initiation complexes[40]. Previous reported binary crystal structures of BmrR–DNA complexes also show a similar but less kinked conformation and unpaired central A:T base pairs of its operator DNA when complexed with a ligand-bound BmrR or a constitutively active derivative of BmrR, indicating that the central kink is mainly attributed to ligand-bound BmrR binding and is further enlarged by RNAP approaching, and also suggesting that such kinked conformation of promoter DNA is required for transcription activation of BmrR (Fig. 2b, c and Supplementary Fig. 5b).

In addition to the central kink (kink 3), we observed three additional kinks at the upstream promoter dsDNA. Kink 1 corresponds to a downward 37° bend of the helical axis at the center of the −35 element (−35); kink 2 corresponds to a upward

52° bend of the helical axis at the middle point (−31) of the DNA region contacted by wing loop and HTH motifs of BmrR$_{US}$; and the kink 4 corresponds to an upward 48° bend of the helical axis at the downstream middle point (−18), the DNA region contacted by the wing loop and HTH motifs of BmrR$_{DS}$ (Fig. 4a). The kink 1 is imposed by σ$^A_4$ and BmrR$_{US}$. The kink 2 and kink 4 are initially imposed by the BmrR dimer as observed in the crystal structure of BmrR–DNA complex, but are further enlarged by σ$^A_4$ and σ$^A_{3.1}$ during engagement of RNAP holoenzyme (Supplementary Fig. 5c–e). Altogether, we propose that ligand-bound BmrR pre-distorts its operator DNA for efficient RNAP recruitment, which further sub-tunes the promoter geometry. The net effect of the four kinks induced by RNAP and BmrR realigns the −35 and −10 regions of promoter dsDNA to the optimal interspace length and geometry to activate transcription from the otherwise silent promoter, which will be discussed in the following section.

**Discussion**

σ$^A_4$ and σ$^A_2$, the two major domains of σ factors, are anchored on the surface of RNAP core enzyme with little flexibility, thereby the RNAP-σ$^A$ holoenzyme could only recognize promoters with a confined length of spacer between the −35 element, recognized by σ$^A_4$, and the −10 element, recognized by σ$^A_2$. Indeed, the majority of bacterial promoter DNAs contain the −35/−10 spacer of 17 ± 1 bp[41].

Recent work from Darst's lab has trapped a near-complete set of intermediate states during RPo formation and for the first time provided a full picture of the complicated process of

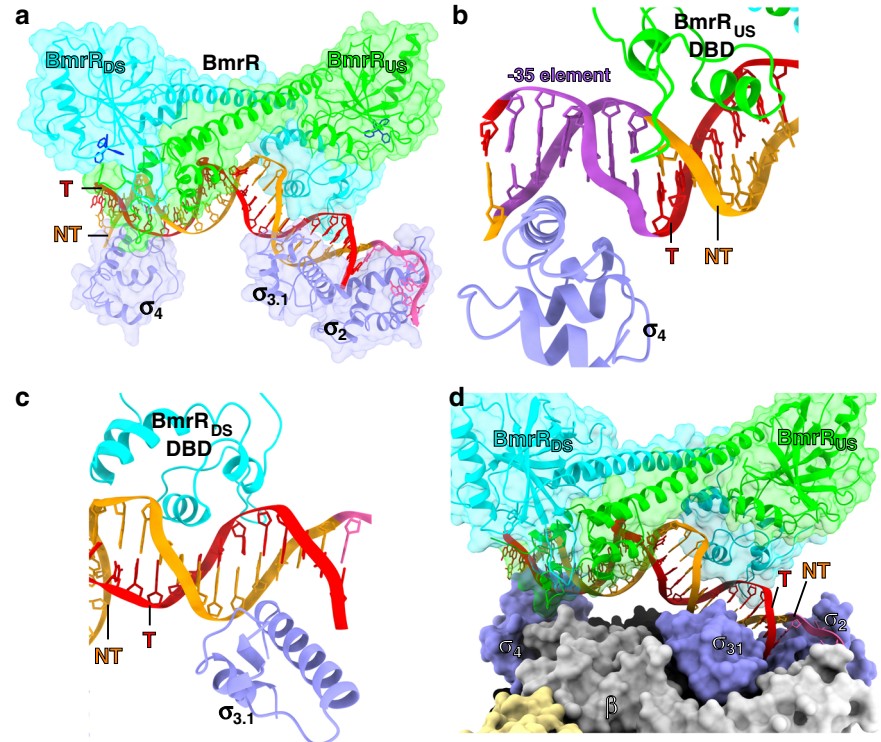

**Fig. 3 Cooperative recognition of promoter DNA by BmrR and RNAP. a** The upstream promoter dsDNA is fully protected by BmrR and $\sigma^A$ (surface and cartoon presentation). **b** BmrR$_{US}$ and $\sigma^A_4$ protects the upstream half of BmrR operator at opposite DNA faces. **c** BmrR$_{DS}$ and $\sigma^A_{3.1}$ protect the downstream half of BmrR operator at opposite DNA faces. **d** BmrR has little contact with RNAP-$\sigma^A$ holoenzyme. The colors are as in Fig. 1.

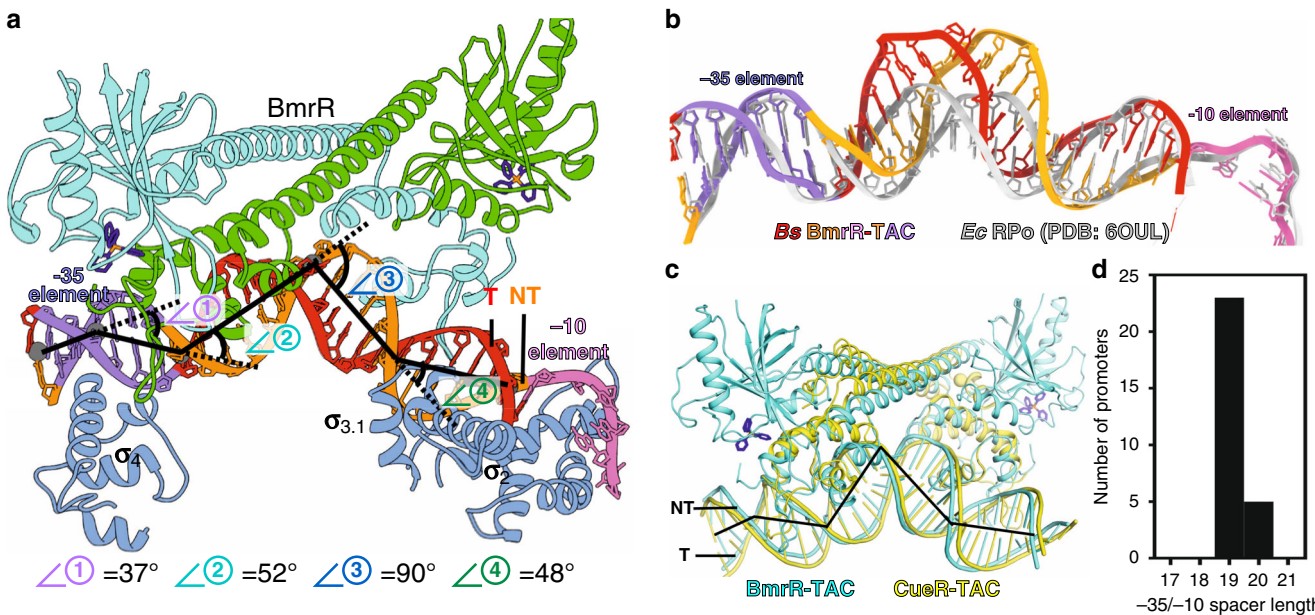

**Fig. 4 BmrR and RNAP induce four kinks of upstream promoter dsDNA. a** The kinks of the upstream promoter DNA at positions −35 (∠①, 37°), −31 (∠②, 52°), −24 (∠③, 90°), and −18 (∠④, 48°). The colors are as in Fig. 1. **b** Comparison of the upstream promoter dsDNA in BmrR-TAC and *E. coli* RPo (PDB: 6OUL). **c** The superimposition of upstream promoter dsDNA in *B. subtilis* BmrR-TAC (cyan) and *E. coli* CueR-TAC (yellow). **d** The distribution of −35/−10 spacer length of MerR-TF-regulated promoters reported in the literature (see Supplementary Table 2 for promoter sequences).

promoter unwinding using an *rps* T2 promoter with a −35/−10 spacer of 17 bp[42]. The structures show that RNAP-$\sigma^{70}$ first recognizes the −35 element in a sequence-dependent manner, places the downstream straight dsDNA on top of the RNAP-β protrusion, and interacts with −10 element in a sequence-independent manner in the RPc. Subsequently, a rotation of

dsDNA at the −10 element precisely places the −12/−11 base pairs near the tryptophan dyad (a key structural motif for separating promoter dsDNA at the upstream junction) and the −11A pocket (another key structural motif for recognizing and trapping base of the first unwound and flipped nucleotide), resulting in the spontaneous unwinding the −11 base pair and

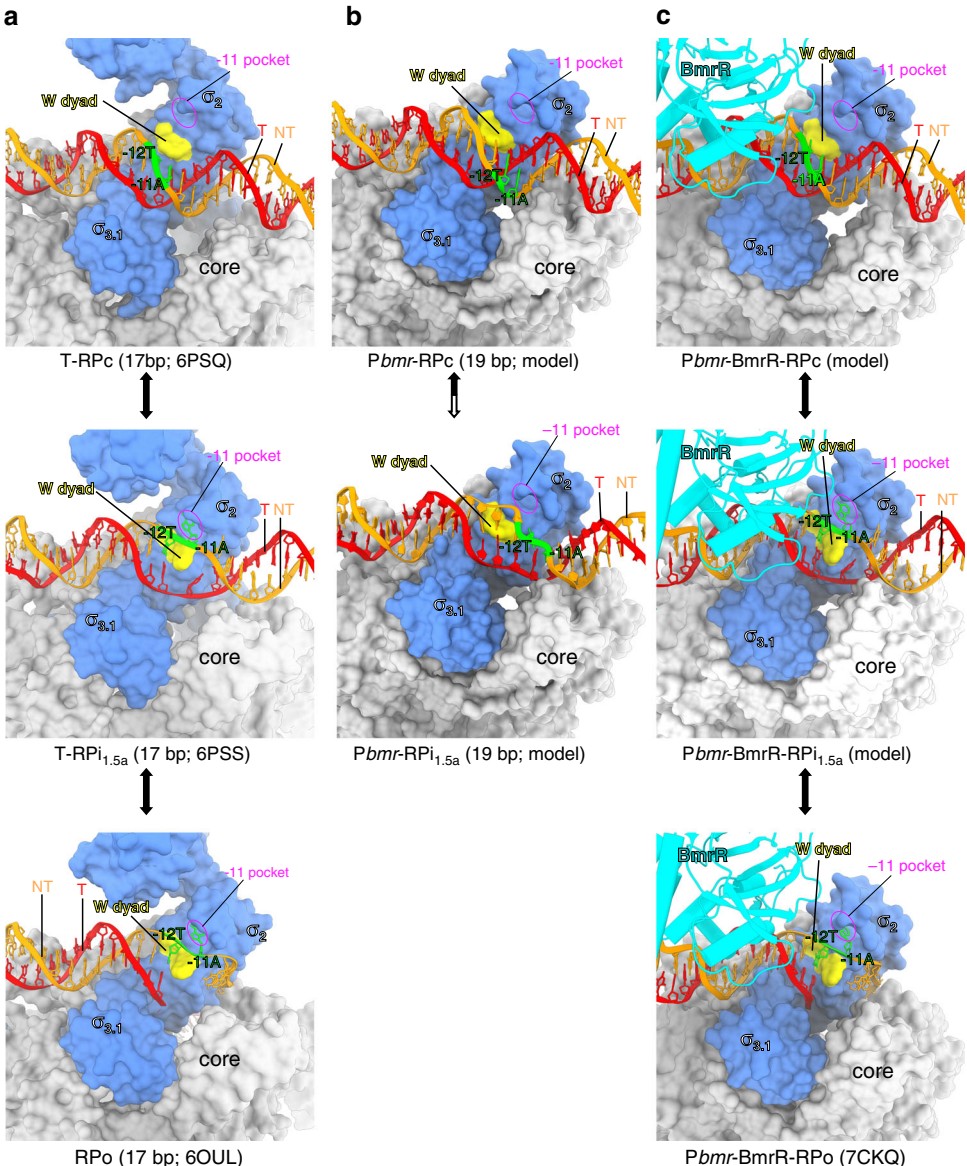

**Fig. 5 The proposed mechanism of transcription activation by BmrR. a** The simplified pathway of *rps* T2 promoter (17-bp −35/−10 spacer) unwinding by *E. coli* RNAP-σ[70] holoezyme. The *Ec* T-RPc (PDB: 6PSQ; top panel), RPi$_{1.5a}$ (PDB: 6PSS; middle panel), and RPo (PDB: 6PSW; bottom panel) were adapted from ref. [42]. **b** The proposed aborted pathway for unwinding P*bmr* by *B. subtilis* RNAP-σ[A] holoezyme. The P*bmr*-RPc (upper panel) was built base on the *Ec* T-RPc (PDB: 6PSQ); the hypothetic intermediate of P*bmr* unwinding by *B. subtilis* RNAP-σ[A] promoter, in which the −12 T:A base pair and −11A is far away from the W-dyad and −11A pocket, two key structural elements for protein unwinding, and the −13T$_{(nt)}$ is not able to be secured by the −11A pocket. **c** The proposed pathway for TPP-BmrR-bound P*bmr* unwinding by *B. subtilis* RNAP-σ[A] holoezyme. The P*bmr*-BmrR-RPc (upper panel) was built base on the *Ec* T-RPc (PDB: 6PSQ); the P*bmr*-BmrR-RPi$_{1.5a}$ (middle panel) was built base on the *Ec* T-RPc (PDB: 6PSQ); and the P*bmr*-BmrR-RPo was determined in this study. Red, template DNA; orange, non-template DNA; blue, σ[A]; cyan, BmrR; yellow patch, W-dyad; purple circle, the −11A pocket.

securing the non-template −11A into its pocket at the stage of RPi$_1$ and RPi$_{1.5}$ (Fig. 5a).

In a scenario of a promoter with a non-optimal −35/ −10 spacer, such as the BmrR-regulated P*bmr* containing a 19-bp spacer (2-bp offset from the optimal 17-bp spacer), we infer that the RNAP-σ[A] is able to locate and bind the −35 element, and subsequently place the straight dsDNA on top of the RNAP-β protrusion as it does for the promoter with a 17-bp spacer. However, at the stage of RPi$_1$ formation, the rotation of promoter is not able to position the −12/−11 base pairs near the two key structural motifs, especially the spontaneously flipped −11A is far from its pocket, resulting in the failure of initiating of promoter unwinding and subsequent quick dissociation of the unstable complex (Fig. 5b).

In the presence of ligand-bound BmrR, although the upstream DNA is highly distorted, the four complex kinks induced by RNAP and BmrR together realign the helical axis of the −35 and −10 regions, allowing the −35 and −10 regions of promoter dsDNA to load on σ$_4$ (in a sequence-specific manner) and the top of RNAP-β protrusion (in a sequence-nonspecific manner), respectively, as in the modeled BmrR-RPc (Fig. 5c). More importantly, the four complex kinks shorten the distance between the −35 and −10 elements, restore the proper stereochemistry of the two elements, position the −12/−11 base pairs near the tryptophan dyad and the −11A pocket ready for the subsequent spontaneous unwinding of the −11 base pair and securing the non-template −11A into −11A pocket in a BmrR-RPi$_{1.5}$ structure

model (Fig. 5c), and finally leads to the formation of a stable BmrR-TAC as determined in the study.

Structure superimposition of the *B. subtilis* BmrR-TAC and *E. coli* CueR-TAC (containing the other prototype member of the MerR-TF family, CueR, that also activates promoters with a 19-bp non-optimal −35/−10 spacer; determined in our parallel study) unveils the same four complex kinks at the upstream promoter dsDNA (Fig. 4c)[43]. On the other hand, a survey of 28 experimentally confirmed promoters regulated by known MerR-family TFs reveals that all promoters contain 19- or 20-bp spacers between the −35 and −10 elements (Fig. 4d), suggesting that the MerR-family TFs probably activate transcription by using the same DNA-distortion and RNAP-non-contact mechanism. The bacterial RNAP-σ[70] holoenzyme itself could accommodate promoter DNA with 17 ± 1 bp spacer by rotating the σ[4] domain anchored on the flexible tip helix of the β flap[44]. Thereby, BmrR probably binds and orients 20-bp spacer DNA in a similar manner as it does for 19-bp spacer DNA and the rotation of σ[4] domain accommodates the additional base pair.

The majority of TFs employ RNAP–protein contacts to recruit RNAP for initiating transcription of downstream genes. Given the significant overlapping binding sites of MerR-TFs and σ factor, it is proposed that transcription activation by MerR-TFs also involves RNAP interaction besides DNA distortion[45]. This study resolves the long-standing question by showing that the MerR-TFs activate transcription solely through DNA distortion involving no interaction with RNAP holoenzyme, and thus support an uncanonical paradigm of RNAP-non-contact transcription activation. This transcription regulation mechanism provides a precedent for a variety of other eukaryotic factors that work by manipulating the local DNA structure to change transcription output, including TATA-binding protein, chromatin remodelers, and, importantly, the nucleosome core particle itself.

## Methods

**Plasmid construction**. DNA fragments encoding *B. subtilis* BmrR and σ[A] were amplified from *B. subtilis* genomic DNA, and cloned into pTolo-EX5 (ToloBio Inc.) or pET-28a, respectively, using homologous recombination (Novoprotein). The pEASY/P*bmr* was constructed by inserting the promoter region (−50 to +50) of *bmr* gene (amplified from *B. subtilis* genomic DNA using primers: forward primer, 5′-TTTGCAAATCCGTTGACTCTCCCC-3′; and reverse primer, 5′-AAATAAAAAGGCCTGCGATTACCAGCAGGCCTTAAGGTAATATTTTTC TTCTCCATATGAC-3′ (with the tR2 terminator sequence underlined)) into the pEASY-blunt vector (Transgen Biotech). The detailed primer sequences are listed in Supplementary Table 3.

**Bacillus subtilis BmrR**. The expression of *Bs*-BmrR was induced by 0.5 mM iso-propyl β-D-1-thiogalactopyranoside (IPTG) at 18 °C for 14 h in *E. coli* BL21(DE3) cells carrying pTolo-EX5-*Bs*-BmrR. The cells were lysed in lysis buffer A (50 mM Tris-HCl, pH 8.0, 0.3 M NaCl, 5% (v/v) glycerol, 5 mM β-mercaptoethanol, protease inhibitor cocktail (Bimake.com Inc.)) using an Avestin EmulsiFlex-C3 cell disrupter (Avestin Inc.). The *Bs*-BmrR were enriched by a gravity column packed with 2 ml Ni-NTA agarose (Smart Life Sciences Inc.), washed with lysis buffer A containing 20 mM imidazole, and eluted with lysis buffer A containing 300 mM imidazole. The eluted fractions were treated with TEV protease and dialyzed overnight in a dialysis buffer (50 mM Tris-HCl, pH 8.0, 0.1 M NaCl, 5% (v/v) glycerol, 5 mM β-mercaptoethanol). The sample was reloaded onto a Ni-NTA column to remove the impurity. The fraction containing *Bs*-BmrR were further loaded onto a Heparin column (HiTrap Heparin HP 5 ml column, GE Healthcare Life Sciences), and eluted with a salt gradient of buffer A (50 mM Tris-HCl, pH 8.0, 0.1 M NaCl, 5% (v/v) glycerol, 1 mM dithiothreitol (DTT)) and buffer B (50 mM Tris-HCl, pH 8.0, 1 M NaCl, 5% (v/v) glycerol, 1 mM DTT). The elute fractions containing target proteins were collected, concentrated to 2 mg/ml, and stored at −80 °C.

**Bacillus subtilis σ[A]**. The *E. coli* BL21(DE3) cells carrying pET-28a-σ[A] were cultured at 37 °C in LB medium to OD$_{600}$ of 0.6–0.8. Protein expression was induced by 0.5 mM IPTG at 18 °C for 14 h. Cells were harvested and pellets were suspended in lysis buffer B (50 mM Tris-HCl, pH 8.0, 0.3 M NaCl, 5% (v/v) glycerol, 5 mM β-mercaptoethanol, protease inhibitor cocktail (Bimake.com Inc.)) and lysed using an Avestin EmulsiFlex-C3 cell disrupter (Avestin Inc.). The supernatant of the lysate was loaded onto a gravity column packed with 2 ml Ni-NTA agarose (Smart

Lifesciences Inc., China), which was washed with lysis buffer B containing 20 mM imidazole and eluted with lysis buffer B containing 300 mM imidazole. The eluted fractions were dialyzed overnight in the dialysis buffer as described above. The sample were loaded onto a Heparin column (HiTrap Heparin HP 5 ml column, GE Healthcare Life Sciences), and eluted with a salt gradient of buffer A (50 mM Tris-HCl, pH 8.0, 0.05 M NaCl, 5% (v/v) glycerol, 1 mM DTT) and buffer B (50 mM Tris-HCl, pH 8.0, 1 M NaCl, 5% (v/v) glycerol, 1 mM DTT). The elute fractions containing *Bs* σ[A] were concentrated to ~4 mg/ml and stored at −80 °C.

**Bacillus subtilis RNAP holoenzyme**. *Bacillus subtilis* RNAP holoenzyme was isolated from *B. subtilis* (strain 168). The cell pellet was resuspended in the lysis buffer C (40 mM Tris-HCl, pH 7.7, 200 mM NaCl, 5% glycerol, 2 mM EDTA, 2 mM DTT, 0.1 mM phenylmethylsulfonyl fluoride (PMSF), and protease inhibitor cocktail (Biomake.com Inc.)) and lysed using a cell disrupter (Avestin EmulsiFlex-C3). The supernatant was precipitated with polyethylenimine (0.7% final) at 4 °C. The pellet was washed and then resuspended with TGED buffer (10 mM Tris-HCl, pH 7.7, 5% glycerol, 1 mM DTT, and 2 mM EDTA) with an additional 1 M NaCl. The supernatant was precipitated by ammonium sulfate (35 g/ml final) and the resulting pellet was collected and dissolved with TGED buffer. The supernatant was loaded onto a Heparin column (HiTrap Heparin HP 5 ml column, GE Healthcare Life Sciences), and eluted with a salt gradient of buffer HPA (40 mM Tris-HCl, pH 8.0, 0.1 M NaCl, 5% (v/v) glycerol, 1 mM DTT, and 0.1 mM EDTA) and buffer HPB (50 mM Tris-HCl, pH 8.0, 1 M NaCl, 5% (v/v) glycerol, 1 mM DTT, and 0.1 mM EDTA). The eluted fractions were further loaded onto a Mono Q column (Mono Q 10/100 GL, GE Healthcare Life Sciences), followed by a salt gradient of buffer QA (40 mM Tris-HCl, pH 7.7, 100 mM NaCl, 5% (v/v) glycerol, 1 mM DTT, 0.1 mM EDTA) and buffer QB (40 mM Tris-HCl, pH 7.7, 600 mM NaCl, 5% (v/v) glycerol, 1 mM DTT, 0.1 mM EDTA). The fractions containing target proteins were collected, concentrated to 5 mg/ml, and stored at −80 °C. The yield of RNAP holoenzyme is 0.4 mg/l.

**Assembly of B. subtilis BmrR-TAC**. Nucleic-acid scaffold for *B. subtilis* BmrR-TAC was prepared from synthetic oligos: non-template-strand DNA (0.5 mM final; Sangon Biotech), template-strand DNA (0.6 mM final; Sangon Biotech), and RNA (0.75 mM final; GenScript Biotech Corp.) in an annealing buffer (5 mM Tris-HCl, pH 8.0, 200 mM NaCl, and 10 mM MgCl$_2$) by an annealing procedure (95 °C, 5 min, followed by 2 °C step cooling to 25 °C).

*Bs* RNAP-σ[A] holoenzyme, *Bs* σ[A], *Bs*-BmrR, and the nucleic-acid scaffold were incubated in a 1:0.5:4:1.3 molar ratio in the presence of 0.5 mM TPP at 4 °C for 1 h (The additional 0.5 molar ratio of *Bs* σ[A] is included in the mixture to convert possible contamination of RNAP core enzyme to *Bs* RNAP-σ[A] holoenzyme.) The mixture was applied to a Superose 6 10/300 GL column (GE Healthcare Life Sciences) equilibrated in 10 mM HEPES, pH 7.5, 100 mM KCl, 5 mM MgCl$_2$, 0.5 mM TPP, and 3 mM DTT. Fractions containing *B. subtilis* BmrR-TAC were collected and concentrated to ~19 mg/ml.

**Cryo-EM structure determination**. The freshly purified *B. subtilis* BmrR-TAC at 15 mg/ml was mixed with 3-([3-cholamidopropyl] dimethylammonio)−2-hydroxy-1-propanesulfonate (CHAPSO, 8 mM final; Hampton Research Inc.) prior to grid preparation. The mixture (3 μl) was loaded on the glow-discharged (120 s) C-flat CF-1.2/1.3 400 mesh holey carbon grids and excess samples were blotted for 8 s with a blot force of −2 at 10 °C and 95% humidity in the chamber of Vitrobot Mark IV (Thermo Fisher Scientific). Subsequently, the grids were vitrified by plunging into liquid ethane.

Cryo-EM imaging was performed on a Titan Krios equipped with a Gatan K2 Summit direct electron detector. Data were collected at a nominal magnification of ×22,500 (1.0 Å/pixel) with a dose rate of 8 electrons/pixel/s on sample (~7.4 electrons/pixel/s on detector). A total of 2226 images were recorded using Serial EM with super mode for 7.6 s exposures in 38 subframes to give a total dose of 60.8 electrons/Å[2] with defocus range of −1.5 to −2.5 μm. Frames in individual movies were aligned using MotionCor2[46], and Contrast-transfer-function estimations were performed using CTFFIND4[47]. Auto-picked 494,295 particles were manually inspected and subjected to 2D classification specifying 100 classes in RELION 3.0[48]. Poorly populated classes were removed, resulting in a dataset of 293,724 particles. We use a 40 Å low-pass-filtered map calculated from the cryo-EM structure of *E. coli* CueR-TAC (PDB: 6LDI) as the starting reference model for 3D classification. Among the 3D classes, the best-resolved class containing 118,891 particles. The dataset of 118,891 particles calculated through 3D auto-refinement, CTF-refinement, Bayesian polishing, and post-processing in RELION 3.0. Subsequent masked 3D classification (*N* = 2, without alignment) was performed on 118,891 particle projection by subtracting density outside the BmrR, upstream DNA, and σ[A]. The best-resolved class containing 103,226 particles and calculated through 3D auto-refinement, post-processing in RELION 3.0. The Gold-standard Fourier-shell-correlation analysis indicated a mean map resolution of 4.4 Å.

The cryo-EM structure of *E. coli* CueR-TAC (PDB: 6LDI), crystal structure of *Bs*-BmrR–DNA (PDB:3Q3D), and crystal structure of *Geobacillus stearothermophilus* YkzG (PDB:4NJC) were manually fit into the cryo-EM density map in Coot[49], followed by adjustment of main- and side-chain conformations in Coot, and real space refined using Phenix[50].

**Fluorescence-detected in vitro transcription assay**. In light of the capability of bulky noncanonical primer-dependent transcription initiation[51], we developed a fluorescence-detected in vitro transcription assay using a 5′ Cy3-labeled CGA RNA primer. *Bacillus subtilis* BmrR-TAC was assembled as above, except that RNA primer was omitted. The reactions mixtures (20 μl) containing 1 μM *B. subtilis* BmrR-TAC and 1 μM 5′ Cy3-labeled CGA RNA were pre-incubated on ice in transcription buffer (40 mM Tris-HCl, pH 8.0, 100 mM NaCl, 10 mM MgCl₂, 12% glycerol, 50 μg/ml bovine serum albumin, 1 mM DTT). The reactions were allowed by the addition of 1 μl 20 μM CTP (final concentration, 1 μM) and further incubation at 37 °C for 30 min and subsequently terminated by the addition of 5 μl stop buffer (8 M urea, 20 mM EDTA, and 0.025% xylene cyanol). The reaction mixtures were heated at 95 °C for 5 min, transferred to ice for 5 min, and electrophoresed on 20% TBE-Urea polyacrylamide gels. The signals were scanned on a fluorescence imager using a 532 nm laser and 570 ± 20 nm filter (Typhoon; GE Healthcare Inc.).

**Radiochemical in vitro transcription assay**. Templates for in vitro transcription assay were amplified by PCR using pEASY/P-*bmr* as a template and M13 primers. Reactions mixtures (20 μl) containing 50 nM *B. subtilis* RNAP holoenzyme, 200 nM σ$^A$, 400 nM BmrR, 200 μM TPP, and 50 nM promoter DNA were pre-incubated at 37 °C for 15 min in transcription buffer. The reactions were allowed by addition of 1 μl NTP mixture to reach a final concentration of 100 μM CTP, 100 μM GTP, 100 μM ATP, 100 μM CTP, and 100 μM [α-32P]UTP (0.04 Bq/fmol) and further incubation at 37 °C for 15 min, and terminated by the addition of 5 μl stop buffer (8 M urea, 20 mM EDTA, 0.025% xylene cyanol, and 0.025% bromophenol blue). The reaction mixtures were heated at 95 °C for 5 min, transferred to ice for 5 min, electrophoresed on 15% TBE-Urea polyacrylamide gels, and analyzed by phosphor imaging (Typhoon; GE Healthcare Inc.).

**DNA structure analysis**. The groove widths were measured with w3DNA (http://web.x3dna.org/), and the data were plotted in SigmaPlot14.0 (Systat Software Inc.). The DNA-distortion angles (kinks) were measured with UCSF Chimera. The centroid positions were created at the base-pair center of sites −38, −35, −31, −24, −18, and −13, and the angles between the centroid positions were measured using the Angles/Torsion function.

**Gel-shift assay**. Nucleic-acid scaffolds (sequences are listed in Supplementary Fig. 1a) for the gel-shift assay were prepared as above. Reaction mixtures contained (4 μl): 0.75 μM *B. subtilis* RNAP holoenzyme, 0.5 μM P*bmr*-2 or P*bmr*-wt1 promoter DNA, 0 or 2 μM BmrR, 0 or 500 μM TPP in 10 mM HEPES pH 7.5, 100 mM KCl, 5 mM MgCl₂, and 3 mM DTT. Reaction mixtures were incubated for 1 h at room temperature and followed by heparin challenge (100 μg/ml; final concentration) when indicated. The complexes were separated by 5% TBE gel in the TBE buffer (90 mM Tris-borate, pH 8.0, and 2 mM EDTA), stained using SYBR-Gold, and analyzed by Tanon-2500 (Tanon Science & Technology Co., Ltd).

**Reporting summary**. Further information on experimental design is available in the Nature Research Reporting Summary linked to this paper.

## Data availability

The final coordinate and cryo-EM map were deposited into the protein data bank and the electron microscopy data bank with accession codes of 7CKQ and 30390, respectively. Other data are available from the corresponding authors upon reasonable request. Source data are provided with this paper.

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

## Acknowledgements
The work was supported by the Strategic Priority Research Program of the CAS to Y.Z. (XDB29020000), the National Key Research and Development Program of China to Y.Z. (2018YFA0900701), the Shanghai Science and technology innovation program to Y.Z. (19JC1415900), the National Natural Science Foundation of China to Y.Z. (31822001 and 31970040) and to Q.L. (81973399), and the Leading Science Key Research Program of CAS to Y.Z. (QYZDB-SSW-SMC005), and the Shanghai "Rising Stars of Medical Talent" Youth Development Program to Q.L. (Youth Medical Talents—Clinical Pharmacist Program), and grants from the National Institutes of Health of the United States to T.V.O. (GM038784-31 and CA193419). We thank Liangliang Kong and Fangfang Wang at the cryo-EM center of the National Center for Protein Science Shanghai (NCPSS) for assistance with data collection.

## Author contributions

L.L. purified endogenous *B. subtilis* BmrR, assembled BmrR-TAC complex, and performed the gel-shift and transcription assays. C.F. collected the cryo-EM data, calculated the cryo-EM maps, and completed the model building. Y. Zhao assisted in complex assembly, X.W. and L.Y. assisted in cryo-EM data collection and structure determination. M.Z., X.S., Q.L., and Y. Zhang designed and supervised the project. T.V.O., S.J.P., Q.L., and Y. Zhang provided mechanistic interpretation. Q.L. and Y. Zhang wrote and edited the manuscript.

## Competing interests
The authors declare no competing interests.
