## [Peer Review File · Nature Communications]

REVIEWER COMMENTS

Reviewer #1 (Remarks to the Author):

Summary

In this manuscript, Fang and colleagues report a 4.4-Å nominal resolution cryo-EM structure of a complex containing *B. subtilis* RNA polymerase, σ^A , promoter DNA, and ligand-bound BmrR transcription factor. An atomic model could be generated using prior structures of transcription activation complexes and BmrR in complex with operator DNA. The results presented here confirm previous BmrR-DNA structural data indicating that BmrR distorts and realigns promoter DNA, thereby activating transcription. The structure shows how DNA is distorted in the context of the RNAP holoenzyme. It additionally shows that BmrR makes little direct contact with the RNAP holoenzyme, supporting an RNAP-contact-independent mechanism. Because of these advances, I support publication of the manuscript after minor revisions as outlined below.

Major Comments

1) p. 6, line 14: The promoter sequences of *B. subtilis* P_{bmr} seem to differ slightly between this study and previous crystallography studies. Does the operator sequence affect the degree of bending? Can the authors make any inferences about how BmrR would bind and reorient DNA in the context of a 20-bp spacer sequence? This is also related to the bending shown in Fig. 4A, and the authors' analogy to TBP.

2) p. 12: The described methods require additional clarification. The authors report that σ^A was cloned into pET-28a, but no purification is described and the assembly protocol doesn't mention adding additional σ^A . Was purified RNAP associated stoichiometrically with σ^A and no other sigma factors? Where is the excess σ^A coming from in the size exclusion chromatogram shown in Fig. S1B?

Minor comments

1) p. 5, line 6: It would be helpful to see an indication in Fig. 1A of which bases are derived from the natural promoter sequence, as opposed to engineered for stability.

2) p. 5, line 7: In Fig. S1A, my understanding is that the "P_{bmr} wild type" promoter is double-stranded DNA, and the "P_{bmr} derivative" contains a mismatched region. Does the "P_{bmr} derivative" also contain RNA?

3) p. 5, line 9: Since Fig. 1 and Fig. S1 are referenced before TPP is defined in the text, describing this compound in the figure legends would be helpful.

4) p. 5, line 13: Not all of Fig. S2 is referenced in the main text.

5) p. 5, line 17: There is no Fig. S3C.

6) p. 7, line 16-17: Should there be an indication of the extended -10 region in Fig. 1A?

7) p. 12, line 5: Should this be homologous recombination?

8) p. 12, line 46: Why is template strand DNA used at a higher concentration than nontemplate strand DNA?

9) p. 13, line 13: Is the reported dose rate the exposure rate on the sample? Could the authors please also report the approximate exposure rate on the detector?

10) Fig. S2D: Was the local resolution calculated using Resmap or some other program?

Reviewer #2 (Remarks to the Author):

This manuscript from Fang et al., titled "The bacterial multidrug resistance regulator BmrR distorts promoter DNA to activate transcription" elucidated the molecular mechanism of transcription activation by bacterial multidrug resistance regulator BmrR by determining the cryo-EM structure of a BmrR bound *B. subtilis* RNA polymerase (RNAP) transcription activation complex (TAC). BmrR belongs to the MerR transcription factor (TF) family, which is noted by its unique transcription activation strategy. Contrast to canonical promoter DNAs that contain 17-bp spacer between -35 and -10 elements, the promoter DNAs dependent on MerR family TFs contain 19-20 bp spacer that is suboptimal for the promoter recognition by transcription initiation factor σ . In response to various stimuli such as metal ions, oxidative stress, and small molecules such as antibiotics, MerR TFs bind to these suboptimal promoters and adjust the physical distance between -35 and -10 elements by distorting the DNA, leading to the transcription activation of the promoters.

Fang et al., find that (1) BmrR does not directly contact to *B. subtilis* RNAP, but recognizing the upstream promoter DNA from the opposite side of the RNAP, inducing steeper kinks than BmrR alone; (2) that the kinked DNA locates the significant residues of the -10 element near the tryptophan dyad, facilitating promoter melting leading to transcription activation; (3) the first high-resolution structure of *B. subtilis* RNAP including its epsilon subunit. Probably with a parallel study on an *E. coli* CueR-bound TAC complex as written in the manuscript, this study expands our knowledge on the molecular mechanism of transcription activation, in particular in response to potentially life-threatening stimuli. From the novelty and mechanistic insights of the study, this reviewer recommends this manuscript to be published in Nature communications after those topics are addressed:

1. This manuscript does not contain any transcription assay result revealing that the reconstituted complex is capable of transcription. To convince that the reconstituted complex is in an active conformation as well as BrmR shows the expected activity, transcription assay needs to be performed.

2. In the abstract, the authors wrote "the structure supports a DNA-distortion and RNAP-contact-independent paradigm of transcription activation by MerR TFs." This statement provides the impression that MerR binding to the promoter is necessary and sufficient for the transcription activation. However, the manuscript revealed that RNAP binding to the promoter DNA enhanced each kink, and this would adjust the distance between -35 element and -10 element significantly. The manuscript also describes the promoter recognition of the RNAP and BmrR as 'cooperative.' Therefore, it would be confusing to use the term 'RNAP-contact-independent'. 'RNAP-non-contact' could be neutral.

3. What would make the four kinks even steeper with RNAP binding? Could you add some postulation for the conformational changes? And how much the changes shorten the linker between -35 and -10 elements compared to the linker without RNAP binding to the DNA? You can add a supplemental figure if necessary.

4. The manuscript addresses that the interaction between -35 and -10 elements and σA is identical to that found in the previously known structures (page 7, line 4). To reveal the point clearer, would you add a figure overlapping the Ca traces of the promoter DNAs and sigma factor region of the *B. subtilis* TAC with those in canonical transcription initiation complexes? Or you can add rmsd

values of the trances. This would reveal how much the linker region of the DNA deviates from the position of canonical promoter DNA as well as how the sigma factor locates with the MerR TFs-dependent promoter with its domains.

Minor revision points:

1. the homology model of the epsilon subunit of *B. subtilis* RNAP used for the modeling is written in the table S1, but not in method section (page 13, line 31)
2. In Page 6, line 20. doesn't  does not
3. In the preparation of endogenous *B. subtilis* RNAP holoenzyme, PEI precipitation and high salt elution were used before ammonium sulfate precipitation, heparin column, and mono Q column chromatography. For *E. coli* RNAP purification, a similar procedure might produce core enzyme without sigma factor. Could you give some explanation of how the procedure retains σ^A for the *B. subtilis* holoenzyme?
4. page 14, line 5. Yellow, non-template DNA  orange color?

Reply to reviewer's comments

Referee #1

Summary

In this manuscript, Fang and colleagues report a 4.4-Å nominal resolution cryo-EM structure of a complex containing *B. subtilis* RNA polymerase, sigmaA, promoter DNA, and ligand-bound BmrR transcription factor. An atomic model could be generated using prior structures of transcription activation complexes and BmrR in complex with operator DNA. The results presented here confirm previous BmrR-DNA structural data indicating that BmrR distorts and realigns promoter DNA, thereby activating transcription. The structure shows how DNA is distorted in the context of the RNAP holoenzyme. It additionally shows that BmrR makes little direct contact with the RNAP holoenzyme, supporting an RNAP-contact-independent mechanism. Because of these advances, I support publication of the manuscript after minor revisions as outlined below.

We thank reviewer #1 for the thoughtful comments and recommendation that this work be published in *Nature Communications* after minor revisions. Please see below reply for specific questions.

Major Comments

1) p. 6, line 14: The promoter sequences of *B. subtilis* P_{bmr} seem to differ slightly between this study and previous crystallography studies.

The *P_{bmr}* used in this study contain a pseudo-palindromic BmrR operator sequence "GACTCTCCCCTAGGAGGAGGTC" that matches exactly to that used in previous crystallography study (PMID:11201751 and 21690368 or refs 7 and 24). Because BmrR dimerization is mediated by the crystallographic two-fold symmetry in the crystal structures of BmrR-DNA complexes, the DNA sequences fit into the electron density were chosen to be symmetric and thus differed from the sequences used for crystallization.

Does the operator sequence affect the degree of bending?

We infer that the operator sequence doesn't affect degree of bending. As described in the manuscript, phosphate backbone interactions make substantial contribution to the BmrR-DNA interface. Moreover, the superimposition of CueR-TAC and BmrR-TAC, which contain different promoter sequences and distinct TFs, shows almost identical degrees of bending (Fig. 4B).

Can the authors make any inferences about how BmrR would bind and reorient DNA in the context of a 20-bp spacer sequence? This is also related to the bending shown in Fig. 4A, and the authors' analogy to TBP.

The following text has been added into the revised manuscript (p. 10, line 25).

'The bacterial RNAP- σ^{70} holoenzyme itself could accommodate a promoter DNA with 17±1 bp spacer by rotating the σ_4 domain anchored on the flexible tip helix of the β flap. Thereby, BmrR probably binds and orients 20-bp spacer DNA in a similar manner as for 19-bp spacer DNA and the rotation of σ_4 domain accommodates the additional base pair.'

2) p. 12: The described methods require additional clarification. The authors report that sigmaA was cloned into pET-28a, but no purification is described, and the assembly protocol doesn't mention adding additional sigmaA. Was purified RNAP associated stoichiometrically with sigmaA and no other sigma factors? Where is the excess sigmaA coming from in the size exclusion chromatogram shown in Fig. S1B?

We supplemented additional σ^A during preparation of the BmrR-TAC. A sub-section for purification of sigma A (see below) and a sentence in the subsection of ‘Assembly of B. subtilis BmrR-TAC’ was added in the revised manuscript. The purified endogenous *Bs* RNAP- σ^A holoenzyme shows that σ^A is probably stoichiometrically associated and no apparent contamination of other σ factors are present (see new supplementary Fig. 1b). However, we still include additional σ^A to convert possible contamination of RNAP core enzyme to *Bs* RNAP- σ^A holoenzyme.

***B. subtilis* σ^A**

The *E. coli* BL21(DE3) cells carrying pET28a- σ^A were cultured at 37 °C in LB medium to OD₆₀₀ of 0.6 to 0.8. Protein expression was induced by 0.5 mM IPTG at 18 °C for 14h. Cells were harvested and pellets were suspended in lysis buffer B (50 mM Tris-HCl pH 8.0, 0.3 M NaCl, 5% (v/v) glycerol, 5 mM β -mercaptoethanol, protease inhibitor cocktail (Bimake.cn Inc.)) and lysed using an Avestin EmulsiFlex-C3 cell disrupter (Avestin, Inc.). The supernatant of the lysate was loaded on to a gravity column packed with 2 ml Ni-NTA agarose (Smart-lifesciences, Inc., China), which was washed with lysis buffer B containing 20 mM imidazole and eluted with lysis buffer B containing 300 mM imidazole. The eluted fractions were dialyzed overnight in the dialysis buffer as described above. The sample were loaded onto a Heparin column (HiTrap Heparin HP 5ml column, GE healthcare Life Sciences), and eluted with a salt gradient of buffer A (50 mM Tris-HCl pH 8.0, 0.05 M NaCl, 5% (v/v) glycerol, 1 mM DTT) and buffer B (50 mM Tris-HCl pH 8.0, 1 M NaCl, 5% (v/v) glycerol, 1 mM DTT). The elute fractions containing *Bs* σ^A were concentrated to ~4 mg/mL and stored at -80 °C’

*‘Bs RNAP- σ^A holoenzyme, *Bs* σ^A , *Bs* BmrR and the nucleic-acid scaffold were incubated in a 1:0.5:4:1.3 molar ratio in the presence of 0.5 mM TPP at 4 °C for 1h (The additional 0.5 molar ratio of *Bs* σ^A is included in the mixture to convert possible contamination of RNAP core enzyme to *Bs* RNAP- σ^A holoenzyme).’*

Minor comments

1) p. 5, line 6: It would be helpful to see an indication in Fig. 1A of which bases are derived from the natural promoter sequence, as opposed to engineered for stability.

We have added a cyan box indicating modified sequence in the new Fig. 1 and added a sentence in the figure legend.

‘red box, sequenced modified for improving complex stability.’

2) p. 5, line 7: In Fig. S1A, my understanding is that the “Pbmr wild type” promoter is double-stranded DNA, and the “Pbmr derivative” contains a mismatched region. Does the “Pbmr derivative” also contain RNA?

Both ‘Pbmr wild-type’ (Pbmr-2) and ‘Pbmr derivative’ (Pbmr-wt1) used for the gel-shift assay is double-stranded DNA. The ‘Pbmr derivative’ doesn’t contain RNA. The promoter sequences used in this study is summarized in a new panel in supplementary Fig. 1a.

3) p. 5, line 9: Since Fig. 1 and Fig. S1 are referenced before TPP is defined in the text, describing this compound in the figure legends would be helpful.

The full name of TPP is included in the legend of Figure 1.

4) p. 5, line 13: Not all of Fig. S2 is referenced in the main text.

The figure citation of supplementary Fig. 2a-c is added into page 5, line 15.

5) p. 5, line 17: There is no Fig. S3C.

Thanks for pointing out the typo. The citation is corrected.

6) p. 7, line 16-17: Should there be an indication of the extended -10 region in Fig. 1A?

An indication of the extended -10 region has been added into new Fig. 1A.

7) p. 12, line 5: Should this be homologous recombination?

Thanks. The typo is corrected.

8) p. 12, line 46: Why is template strand DNA used at a higher concentration than nontemplate strand DNA?

The non-template ssDNA has much higher affinity towards RNAP due to the sequence-specific recognition of the -10 element, so we included slight excess of template ssDNA to avoid presence of non-template ssDNA in the nucleic-acid scaffold.

9) p. 13, line 13: Is the reported dose rate the exposure rate on the sample? Could the authors please also report the approximate exposure rate on the detector?

Yes. The approximate dose rate on the detector was estimated as 7.4 electrons/pixel/s compared to 8 electrons/pixel/s on sample, which has been included in the method section.

10) Fig. S2D: Was the local resolution calculated using Resmap or some other program?

Yes, the local resolution was calculated using Resmap, which has been added into the figure legend.

Reviewer #2 (Remarks to the Author):

This manuscript from Fang et al., titled “The bacterial multidrug resistance regulator BmrR distorts promoter DNA to activate transcription” elucidated the molecular mechanism of transcription activation by bacterial multidrug resistance regulator BmrR by determining the cryo-EM structure of a BmrR bound *B. subtilis* RNA polymerase (RNAP) transcription activation complex (TAC). BmrR belongs to the MerR transcription factor (TF) family, which is noted by its unique transcription activation strategy. Contrast to canonical promoter DNAs that contain 17-bp spacer between -35 and -10 elements, the promoter DNAs dependent on MerR family TFs contain 19-20 bp spacer that is suboptimal for the promoter recognition by transcription initiation factor σ . In response to various stimuli such as metal ions, oxidative stress, and small molecules such as antibiotics, MerR TFs bind to these suboptimal promoters and adjust the physical distance between -35 and -10 elements by distorting the DNA, leading to the transcription activation of the promoters.

Fang et al., find that (1) BmrR does not directly contact to *B. subtilis* RNAP, but recognizing the upstream promoter DNA from the opposite side of the RNAP, inducing steeper kinks than BmrR alone; (2) that the kinked DNA locates the significant residues of the -10 element near the tryptophan dyad, facilitating promoter melting leading to transcription activation; (3) the first high-resolution structure of *B. subtilis* RNAP including its epsilon subunit. Probably with a parallel study on an *E. coli* CueR-bound TAC complex as written in the manuscript, this study expands our knowledge on the molecular mechanism of transcription activation, in particular in response to potentially life-threatening stimuli. From the novelty and mechanistic insights of the study, this reviewer recommends this manuscript to be published in Nature communications after those topics are addressed:

We thank the reviewer for insightful comments. Our response to specific questions is listed below.

1. This manuscript does not contain any transcription assay result revealing that the reconstituted complex is capable of transcription. To convince that the reconstituted complex is in an active conformation as well as BrmR shows the expected activity, transcription assay needs to be performed.

Thanks for the suggestion. Previous reports have demonstrated that ligand-free and ligand-bound BmrR turned the inactive *Pbmr* to an active promoter (PMID: 18658145 or ref 29). Our radiochemical *in vitro* transcription experiment confirmed previous observation when we started this project a year ago (see new supplementary fig. 1d). However, the COVID-19 epidemic has prevented us from purchasing radiochemical nucleotides for several months, and therefore, preventing us from testing the catalytic activity of the reconstituted BmrR-TAC. As requested by reviewer, to validate the active conformation of the cryo-EM sample of BmrR-TAC, we developed a fluorescence-detected *in vitro* transcription assay for measuring primer-dependent transcription initiation (see Method section). The result showed that the cryo-EM sample was able to incorporate CTP to convert the 5' cy3-labeled trinucleotide (cy3-CGA) to a quadra-nucleotide (cy5-CGAC), confirming it is a catalytically competent complex (supplementary Fig. 1h).

2. In the abstract, the authors wrote “the structure supports a DNA-distortion and RNAP-contact-independent paradigm of transcription activation by MerR TFs.” This statement provides the impression that MerR binding to the promoter is necessary and sufficient for the transcription activation. However, the manuscript revealed that RNAP binding to the promoter DNA enhanced each kink, and this would adjust the distance between -35 element and -10 element significantly. The manuscript also describes the

promoter recognition of the RNAP and BmrR as ‘cooperative.’ Therefore, it would be confusing to use the term ‘RNAP-contact-independent’. ‘RNAP-non-contact’ could be neutral.

Thanks for the suggestion. We have changed the term ‘RNAP-contact-independent’ to ‘RNAP-non-contact’.

3. What would make the four kinks even steeper with RNAP binding? Could you add some postulation for the conformational changes? And how much the changes shorten the linker between -35 and -10 elements compared to the linker without RNAP binding to the DNA? You can add a supplemental figure if necessary.

We think σ_4 and $\sigma_{3.1}$ are responsible for enlarged kinks upon RNAP binding. Modeling the BmrR-bound dsDNA on the RNAP- σ A holoenzyme suggested that the conformation BmrR-bound dsDNA has to undergo further subtune (*i.e.* steeper kinks) to simultaneously engage domains σ_4 and $\sigma_{3.1}$ for subsequent RPo formation. The distances the -35 and -10 elements in the two structures (BmrR-TAC and BmrD-DNA) are similar but the dsDNA path of the -10 end substantially differs when the two structures were superimposed based on the -35 elements. Additional three panels were added into supplementary Fig. 5 (supplementary Fig. 5c-e) to illustrate the difference. The following sentence was added into the revised manuscript (p.8, line 25),

‘The kink 2 and kink 4 are initially imposed by the BmrR dimer as observed in the crystal structure of BmrR-DNA complex but are further enlarged by σ^{\wedge}_4 and $\sigma^{\wedge}_{3.1}$ during engagement of RNAP holoenzyme (Supplementary Fig. 5c-5e)’

4. The manuscript addresses that the interaction between -35 and -10 elements and σ A is identical to that found in the previously known structures (page 7, line 4). To reveal the point clearer, would you add a figure overlapping the Ca traces of the promoter DNAs and sigma factor region of the B. subtilis TAC with those in canonical transcription initiation complexes? Or you can add rmsd values of the trances. This would reveal how much the linker region of the DNA deviates from the position of canonical promoter DNA as well as how the sigma factor locates with the MerR TFs-dependent promoter with its domains.

Thanks for the suggestion. We have added one panel to Fig. 4 (Fig. 4b) and two panels to supplementary Fig. 4 (Supplementary Fig. 4e and 4f) to illustrate the comparison of BmrR-TAC and *E. coli* RPo. Supplementary Fig. 4d compares that -35/ σ_4 interactions of the two structures; supplementary Fig. 4e compares the -10/ σ_2 interaction of the two structures; and supplementary Fig. 4f, superimposed based on σ_4 and σ_2 , compares the positions of σ_2 and σ_4 on RNAP core enzyme (r.m.s.d. values of 1.95 Å for all σ_2/σ_4 C α atoms) as well as the -35/-10 spacer conformations of the two structures.

Minor revision points:

1. the homology model of the epsilon subunit of B. subtilis RNAP used for the modeling is written in the table S1, but not in method section (page 13, line 31)

The homology models were added into the method section.

2. In Page 6, line 20. doesn’t  does not

Thanks. It is corrected.

3. In the preparation of endogenous B. subtilis RNAP holoenzyme, PEI precipitation and high salt elution were used before ammonium sulfate precipitation, heparin column, and mono Q column chromatography. For E. coli RNAP purification, a similar

procedure might produce core enzyme without sigma factor. Could you give some explanation of how the procedure retains σ A for the *B. subtilis* holoenzyme?

We have tried purification of endogenous RNAP from different bacterial species. The PEI precipitation step did prevent us from obtaining RNAP holoenzyme in *T. thermophilus*, but we were able to obtain RNAP holoenzyme in *Caulobacter crescentus*, *Xanthomonas oryzae* pv. *Oryzae*, as well as in *Bacillus subtilis*. I guess RNAP holoenzymes from different bacterial species vary in salt sensitivity.

4. page 14, line 5. Yellow, non-template DNA  orange color?

Thanks. The typo is corrected.

REVIEWER COMMENTS

Reviewer #2 (Remarks to the Author):

The authors provided evidences that (1) The purified *B.subtilis* RNAP and BmrR make a stable open complex with wild type P_{bmr} promoter (P_{bmr}-wt1) and the modified P_{bmr} promoter containing consensus -10 element (P_{bmr}-2) (Supplementary Fig. 1d), (2) the complex is capable of RNA synthesis in the presence of TPP with wild type P_{bmr} promoter containing tR2 terminator (P_{bmr}-wt2) (Supplementary Fig. 1c), and (3) the complex containing the modified P_{bmr} promoter (P_{bmr}-1) used for cryo-EM sample preparation can add one nucleotide (CTP) to 5'-Cy3-labelled CGA RNA oligo (added in the revised manuscript). Although the added experimental data improved the quality and solidity of their story, there are a few questions remained unclear.

1. The fluorescence-labelled transcription assay has been performed in a somewhat convoluted way. The reaction time was 30 minutes which is quite long and the concentration CTP for one CMP extension was 1 μ M which is very low compared to the conventional radiochemical transcription assay condition. For example, even in the current manuscript, the radiochemical in vitro transcription assay is done in 100 μ M NTP concentration for 15 minutes to synthesize 124 nt transcript. In addition, Cy3 dye used for fluorescence labelling is bulkier compared to a nucleotide, so it might affect the structure of the RNAP as well as its activity. Thus, the authors should provide proper rationale of the developed assay conditions. Also, this reviewer is wondering if 12% glycerol is a critical condition for the transcription reaction as well.

2. The manuscript mentioned that the P_{bmr} -10 element was replaced with consensus sequence "to improve the sample homogeneity and enhance protein-DNA interaction" (page 4, line 28). Have you seen any difference in transcriptional activity and initiation efficiency between consensus and original -10 element sequences?

With clear answers to the comments above, the manuscript should be suitable for publication in Nature Communications.

Reply to reviewer's comments

Reviewer #2 (Remarks to the Author):

1. The fluorescence-labelled transcription assay has been performed in a somewhat convoluted way. The reaction time was 30 minutes which is quite long and the concentration CTP for one CMP extension was 1 μ M which is very low compared to the conventional radiochemical transcription assay condition. For example, even in the current manuscript, the radiochemical *in vitro* transcription assay is done in 100 μ M NTP concentration for 15 minutes to synthesize 124 nt transcript. In addition, Cy3 dye used for fluorescence labelling is bulkier compared to a nucleotide, so it might affect the structure of the RNAP as well as its activity. Thus, the authors should provide proper rationale of the developed assay conditions. Also, this reviewer is wondering if 12% glycerol is a critical condition for the transcription reaction as well.

Reply: Thanks for the comment. Please see below explanation for the experimental parameters.

In regard to CTP concentration, the final concentration of 1 μ M was chosen in the fluorescence-labelled transcription assay due to misincorporation activity of the tested higher CTP concentrations (10 and 100 μ M).

In regard to the reaction time, the reaction time of 30 minute was chosen to ensure maximal conversion of the substrate for subsequent detection of reaction products by electrophoresis and LC/MS/MS.

In regard to the Cy3-labeled primer, as multiple-subunit RNA polymerases have intrinsic capability of using natural bulky noncanonical primers (NAD, FAD, dephospho-CoA) to initiate transcription, they should be able to use synthetic primers with a similarly bulky moiety at the 5' terminus (*i.e.* the 5' Cy3 dye) to initiate transcription. Our results confirmed it.

In regard to the glycerol concentration, transcription buffers containing various concentrations of glycerol were reported in literatures. Our lab routinely uses the buffer containing 12.5 % glycerol, but we have also tried different concentrations (even 0% glycerol) for specific cases, which showed variable transcription activities. We think glycerol is not critical for *in vitro* transcription, but its concentration does affect the transcription activity.

We have added a few words in the 'Methods' section as suggested by the reviewer in the revised manuscript.

“In light of the capability of bulky noncanonical primer-dependent transcription initiation⁵⁰, we developed a fluorescence-detected *in vitro* transcription assay using a 5' Cy3-labeled CGA RNA primer”

2. The manuscript mentioned that the P_{bmr} -10 element was replaced with consensus sequence “to improve the sample homogeneity and enhance protein-DNA interaction” (page 4, line 28). Have you seen any difference in transcriptional activity and initiation efficiency between consensus and original -10 element sequences?

Reply: The Supplementary fig. 1d shows wild-type and derivative of P_{bmr} exhibit comparable efficiency of RPo formation.